# Challenges and Opportunities in Antimicrobial Stewardship among Hematopoietic Stem Cell Transplant and Oncology Patients

**DOI:** 10.3390/antibiotics12030592

**Published:** 2023-03-16

**Authors:** Anjali Majumdar, Mansi R. Shah, Jiyeon J. Park, Navaneeth Narayanan, Keith S. Kaye, Pinki J. Bhatt

**Affiliations:** 1Division of Allergy and Infectious Disease, Department of Medicine, Rutgers-Robert Wood Johnson Medical School, New Brunswick, NJ 08901, USA; 2Division of Blood Disorders, Rutgers Cancer Institute of New Jersey, New Brunswick, NJ 08901, USA; 3Englewood Health, Englewood, NJ 07631, USA; 4Rutgers-Ernest Mario School of Pharmacy, Piscataway, NJ 08854, USA

**Keywords:** antimicrobial stewardship, antibacterial prophylaxis, hematopoietic stem cell transplant, hematologic malignancy

## Abstract

Antimicrobial stewardship programs play a critical role in optimizing the use of antimicrobials against pathogens in the era of growing multi-drug resistance. However, implementation of antimicrobial stewardship programs among the hematopoietic stem cell transplant and oncology populations has posed challenges due to multiple risk factors in the host populations and the infections that affect them. The consideration of underlying immunosuppression and a higher risk for poor outcomes have shaped therapeutic decisions for these patients. In this multidisciplinary perspective piece, we provide a summary of the current landscape of antimicrobial stewardship, unique challenges, and opportunities for unmet needs in these patient populations.

## 1. Introduction

A report from *The Lancet* estimated that 4.95 million deaths globally were associated with antimicrobial resistance (AMR) in 2019 [1]. The formation of antimicrobial stewardship programs (ASPs) has been a crucial step in reducing the growing problem of antimicrobial resistance and poor patient outcomes through the optimization of antimicrobial use. The implementation of ASPs has gained more global recognition with the development of guidelines and has become a requirement for accredited hospitals by the Joint Commission in the United States [2,3]. It has been noted that the implementation of ASPs in transplant populations, including in solid organ transplant (SOT) recipients, poses challenges, such as increased susceptibility to infections and prolonged courses of antimicrobials [4,5]. However, no consensus guidelines currently exist for ASP implementation in hematopoietic stem cell transplant (HSCT) patients and patients living with malignancies [6,7].

In this perspective article, we provide a summary of the current landscape of ASP practices, unique challenges, and opportunities for unmet needs in the HSCT and oncology populations.

## 2. Unique Challenges to Antimicrobial Stewardship in Oncology Patients

There are multiple facets involved in the care of patients living with malignancies, which require unique approaches by antimicrobial stewardship. Here we highlight some host and treatment factors that affect antimicrobial use in the immunocompromised host living with malignancy.

### 2.1. Understanding Underlying Host Immune Status and Infectious Risks

There is substantial variability in immune status among oncology patients. The avoidance of immune surveillance, escape from immune destruction, and tumor-promoting inflammation, are some main attributes of cancer [8]. Observed classes of inflammatory and immune response are determined by the characteristics of the originating tissue, proinflammatory mediators released by the tumor or stromal milieu, nature of tissue damage, or the organism associated with carcinogenesis [8]. The increased susceptibility of patients with cancer to infections depends on the status of the malignancy, the dominant type of immune deficiency, and the duration and intensity of therapy-mediated immunosuppression.

Cancers may be related to a chronic infection, or inflammation related to an infection, such as with Epstein–Barr virus, *Helicobacter pylori,* Hepatitis B and C viruses, Human Immunodeficiency Virus (HIV), Human Papillomavirus (HPV), and Human T-Lymphotropic Virus (HTLV). Other malignancies may lead to altered intestinal composition, tissue injury, or physical interactions with hematopoietic cells, leading to an increased vulnerability to infectious complications [9]. The disruption of hematopoiesis with resultant neutropenia in patients with acute leukemias predisposes patients to serious bacterial, fungal, and viral infections [9,10]. A lack of splenic and humoral immunity in patients with B-cell disorders can also cause functional hypogammaglobulinemia [11].

The receipt of chemotherapeutic agents (e.g., cytotoxic, lymphocyte-depleting agents) can cause myelosuppression, impacting infection risk depending on the degree and duration of neutropenia and lymphopenia. Lower bone marrow reserve due to older age, the presence of multiple comorbidities, and poor performance status can propagate the risk of chemotherapy-induced neutropenia [12]. Many patients living with malignancies experience a disruption of mucosal barriers due to radiation and chemotherapy, predisposing them to mucositis, sinusitis, and bacterial translocation, in turn leading to bloodstream infections and neutropenic enterocolitis. Modern therapies, including small/signaling molecule inhibitors, monoclonal and bispecific antibodies, T-cell redirecting cellular therapies, and immune checkpoint inhibitors, can cause unique immune dysfunction, increasing the risk for opportunistic infections such as viral reactivation [13,14,15]. Reviews and case reports pertaining to infectious complications with these newer agents are increasingly being published, revealing that associated complications have yet to be reported regarding these therapies [16,17,18].

### 2.2. Additional Healthcare Considerations and Hematopoietic Stem Cell Transplantation 

Besides inherent immune dysfunction resulting from an underlying malignancy and its treatment, patients living with malignancies are at risk for healthcare-associated infections. Patients within this population may also have indwelling devices, such as Ommaya reservoirs, central venous catheters, and urinary catheters, placing them at risk for device-related infections and complications [19]. Frequent exposure to the healthcare system can also predispose patients to acquisition and colonization with multi-drug resistant organisms (MDRO) [20,21,22].

Recipients of HSCT are predisposed to opportunistic infections due to the nature of conditioning regimens and their effects on the immune system, the disruption of hematopoiesis, the donor type and transplant graft source, the presence of mucosal injury, and the need for hospitalization. As highlighted in Figure 1 which is adapted from Tomblyn et al., the most frequently encountered infections in the pre-engraftment phase include bacterial infections related to gastrointestinal translocation, healthcare-associated infections, and *Candida* spp. infections, followed by infections from cytomegalovirus (CMV), *Aspergillus* spp., and *Pneumocystis* in the early post-engraftment period [23]. Figure 1 also provides examples of prevention strategies used to decrease the incidence of these infections which will be discussed later throughout this piece.

Different transplant graft sources (e.g., autologous stem cells collected from bone marrow or peripheral blood stem cells (PBSC); allogeneic stem cells collected from bone marrow, PBSC, or umbilical cord blood) impact the risk of the development of infections. In the month following an allogeneic bone marrow transplant (BMT), the incidence of infections is 47.9%, compared to 32.8% in allogeneic PBSC transplant, which is possibly related to quicker engraftment in the latter population [25,26,27]. Comparatively, umbilical cord blood transplantations carry the highest risk of infections during the first 100 days after transplant, possibly due to delayed engraftment [27]. The impact of donor type (e.g., matched–related, matched–unrelated) and the use of prophylaxis against graft-versus-host disease (GVHD) upon infectious risk remains unclear [28]. Due to the high probability of GVHD with allogeneic PBSCtransplant, there is an increased incidence of late fungal and viral infections when compared with allogeneic BMT [26]. Up to 20–80% of HSCT patients are affected by acute GVHD which may require heightened immunosuppression, placing patients at further risk for opportunistic and fungal infections [11,29,30].

### 2.3. Drug–Drug Interactions Affecting Antimicrobial Use

Significant drug–drug interactions (DDI) can affect the use of antimicrobials with concomitant chemotherapeutic and adjunctive treatments. For example, triazole agents, such as voriconazole, posaconazole, and fluconazole, are noted to cause QTc prolongation. Pharmacodynamically, triazole agents, fluoroquinolones, and macrolides can enhance the QTc-prolonging effects of antiemetics used to manage chemotherapy-induced nausea and vomiting, such as 5-HT3 receptor antagonists (e.g., ondansetron, granisetron) and dopamine receptor antagonists (e.g., metoclopramide) [31].

Antimicrobials that inhibit or induce the CYP3A4 enzyme can lead to increased or decreased levels of chemotherapeutic agents. For example, triazole agents are CYP3A4 inhibitors, which can affect concentrations of chemotherapeutic agents, such as taxane, vinca alkaloids, busulfan, etoposide, ifosfamide, cyclophosphamide, and tyrosine kinase inhibitors [32,33,34]. The CYP3A4-inhibiting antimicrobials can also complicate the management of serum concentrations of drugs with a narrow therapeutic index (e.g., tacrolimus, cyclosporine, and sirolimus): a notable example is voriconazole, which increases the AUC of sirolimus by 1000%, requiring at least a 90% dose reduction of sirolimus [34]. On the other hand, CYP3A4 inducers, such as rifampin, can decrease the serum concentrations of CYP3A4 substrates; therefore, the interacting medications may need to be dose-adjusted or switched to an alternative [34].

HSCT patients are particularly susceptible to significant drug–drug interactions, due to the frequent need for antimicrobial prophylaxis and treatment, polypharmacy, and the use of immunosuppressants [33]. For example, busulfan, a commonly used alkylating agent in stem cell transplant, can interact with triazoles and metronidazole. Voriconazole, a strong CYP3A4 inhibitor, can reduce busulfan clearance and increase its serum level, necessitating the close monitoring of patients regarding the occurrence of busulfan toxicity. Metronidazole also inhibits CYP3A4 in addition to competing for glutathione, which can increase busulfan levels; thus, metronidazole should be avoided for 72 h before and after busulfan administration [34]. Other antimicrobials that are CYP3A4 inhibitors and can cause DDI with immunosuppressants include macrolides and antiviral agents containing ritonavir (e.g., nirmatrelvir/ritonavir).

It is important to note these interactions as they may impact the therapeutic efficacy of antimicrobials, chemotherapeutic agents, and transplant-related medications. These examples also highlight a need for studies examining the clinical impact of DDI, alternative dosing strategies for mitigation of adverse effects, and development of antimicrobials with fewer interactions.

## 3. Unmet Needs and Opportunities for Antimicrobial Stewardship

Due to an increased susceptibility to infection, strategies such as antimicrobial prophylaxis and early empiric therapy have been studied with respect to reducing infection-related morbidity and mortality. However, the long-term consequences of these strategies can be serious. In this section, we review unmet needs, opportunities for further research, and strategies to improve antimicrobial stewardship in these immunocompromised populations.

### 3.1. Fever and Neutropenia

Approximately 80% of patients with hematologic malignancies and up to 50% of patients with solid tumor malignancies develop fever with underlying neutropenia, triggering the use of empiric antimicrobials to reduce morbidity [35]. Of those with febrile neutropenia, up to 45–50% of cases are due to unexplained fevers with no underlying microbiological cause [36]. The swift resolution of neutropenia facilitates the discontinuation of antibacterial agents for most patients with solid tumor malignancies, but antibacterial discontinuation remains an area of variability among patients with hematologic malignancies, due, in part, to the longer durations of neutropenia observed in these patients [37].

The extended use of empiric antimicrobials can lead to the selection of resistant organisms. For example, the prolonged use of vancomycin can result in increased rates of vancomycin-resistant enterococci infection [11]. Multiple studies evaluating bloodstream infections in patients with malignancies have revealed an increased incidence of extended spectrum beta-lactamase (ESBL) and carbapenem-resistant Enterobacterales (CRE), as well as increased rates of *Pseudomonas aeruginosa* and *Acinetobacter baumanii* infections [38,39]. Shorter durations of therapy have been studied in order to decrease bacterial selective pressure, and the implications of this intervention on morbidity are important points to consider. The ANTIBIOSTOP and the How Long studies have demonstrated that empiric antibacterials can be discontinued in hemodynamically stable neutropenic patients with a resolution of fever and lack of active infection; however, it should be noted that some high-risk patients such as those with GVHD were excluded from these studies [40,41]. This approach regarding the early discontinuation of empiric antibacterials has been recommended by The American Society of Transplantation and Cellular Therapy (ASTCT) and the European Conference on Infections in Leukemia (ECIL) [42,43]. Local centers may consider the development of collaborative guidelines that can account for the ongoing monitoring of patients with febrile neutropenia to streamline empiric antibacterial use.

The targeted use of empiric antimicrobials, such as the use of vancomycin, has been addressed in sources such as the National Comprehensive Cancer Network^®^ (NCCN^®^) guidelines, in efforts to decrease antibacterial resistance [11]. A targeted review of epidemiology and resistance patterns among hospital units and/or among patients with underlying malignancy diagnosis, paired with rapid diagnostic tests (RDT), can help institutions evaluate their empiric antimicrobial choices, when to consider discontinuing antimicrobials, and when to escalate therapy in order to optimize therapeutic efficacy.

### 3.2. Antibacterial Prophylaxis

Antibacterial prophylaxis has been shown to decrease the incidence of Gram-negative infections in patients living with malignancy [11,44]. NCCN guidelines recommend the consideration of fluoroquinolone prophylaxis for neutropenic patients at an intermediate to a highrisk of infection, such as for patients with anticipated neutropenia longer than 10 days, or patients undergoing therapy for acute leukemia [11]. However, fluoroquinolone use is associated with increased rates of *Clostridioides difficile* (*C. difficile*) and may lead to the selection of resistant bacterial pathogens, notably decreasing their effectiveness in patients who are colonized with fluoroquinolone-resistant organisms [45,46]. Additionally, if a patient is receiving fluoroquinolone prophylaxis and develops fevers, empiric antibacterial treatment should not include fluoroquinolone and should entail the use of anti-pseudomonal β-lactams, such as piperacillin/tazobactam, cefepime, or meropenem [46,47,48].

A local data review of resistance patterns for MDRO can help centers create guidelines for the optimal use of fluoroquinolone prophylaxis. For patients with fluoroquinolone allergies, alternative agents, such as oral third-generation cephalosporins and trimethoprim/sulfamethoxazole, can be used [11]. Trimethoprim/sulfamethoxazole (TMP/SMX) prophylaxis has been shown to have similar benefits in terms of mortality when compared to fluoroquinolone prophylaxis; however TMP/SMX has been associated with more adverse effects [49]. In addition, TMP/SMX can compound the risk of leukopenia in patients recovering from chemotherapy. As discussed, it is worthwhile to consider the benefits and risks of antibacterial prophylaxis regimens that have potential adverse events such as antimicrobial resistance.

### 3.3. Antifungal Use and Prophylaxis 

Patients undergoing cytotoxic chemotherapy and HSCT are at an increased risk for invasive fungal infections (IFI), with mortality reaching up to 30% [50,51,52,53]. There is considerable variability in the presentation of IFI, with concurrent thrombocytopenia often limiting the potential benefit of diagnostic bronchoscopy, leaving providers with imaging and indirect serum tests for diagnosis, such as *Aspergillus* galactomannan or beta-D-glucan testing [54,55]. The consensus group of the European Organization for Research and Treatment of Cancer/Invasive Fungal Infections Cooperative Group (EORTC), as well as the National Institute of Allergy and Infectious Diseases Mycoses Study Group (MSG), revised definitions to provide guidance about high-risk populations, imaging findings, and laboratory testing methods, needed to diagnose proven, probable, and possible IFI [54]. These guidelines have helped to standardize the diagnosis of IFI in patients with hematologic malignancies and note the subtleties in various indirect diagnostic methods [54].

Two main strategies have been utilized to treat IFI: a preemptive approach involving the use of diagnostic testing to guide treatment versus empiric antifungal therapy, where antifungal therapy is initiated while awaiting work-up. A recent Cochrane review concluded that pre-emptive antifungal treatment may reduce the duration of antifungal use without affecting IFI-related mortality in patients with febrile neutropenia when compared to empiric antifungal therapy, highlighting a need to study at-risk target populations and approaches to antifungal treatment [56]. The EORTC/MSG guidelines note that the T2Candida panel^®^, approved by the United States Food and Drug Administration (FDA) for the detection of common *Candida* species from whole-blood specimens, has a high negative predictive value, which can also aid ASPs in decreasing antifungal use [54].

Antifungal prophylaxis during periods of high-risk neutropenia and following allogeneic HSCT has now become an integral part of national guidelines and clinical care practices in interest of decreasing IFI incidence [11,57]. The NCCN and ECIL recommend posaconazole for antifungal prophylaxis in patients undergoing treatment for acute myeloid leukemia (AML) and myelodysplastic syndrome (MDS) [11,57]. Both organizations also note the increased risk of IFI in allogeneic HSCT patients, providing similar grading recommendations for the use of posaconazole, voriconazole, and amphotericin formulations (including NCCN grade 2B recommendation for isavuconazole), with the understanding that more research is needed to further support these gradings. For patients with significant DDI with azole agents, some centers have adopted the use of echinocandins, although there is a loss of anti-mold coverage when using this strategy [11,57]. Further studies are needed evaluating the use of antifungal prophylaxis strategies and its effect on IFI incidence as the primary endpoint.

It is estimated that up to 57% of antifungals prescribed are not optimal in many cancer patients due to issues such as inappropriate agent selection and dosing frequency [58]. Although antifungal stewardship is encompassed within antimicrobial stewardship, few consensus guidelines exist to support antifungal stewardship programs [58]. The Mycoses Study Group Education and Research Consortium highlights this need and offers recommendations specific to antifungal use based on the core elements of ASP as described by the Centers for Disease Control and Prevention (CDC) [59,60]. Other suggested strategies for improvement in antifungal stewardship have been published, including local education and guideline development, improvements in diagnostics for identification and susceptibility testing, optimizing antifungal dosing, and multidisciplinary team involvement [61,62].

### 3.4. Antiviral Use and Prophylaxis

Similar to patients undergoing SOT, HSCT patients with cytomegalovirus (CMV) infection have poor outcomes [63,64]. In the past, CMV viremia occurring post-HSCT was largely addressed through a pre-emptive approach, with viral load monitoring and corresponding treatment initiation with the antiviral agents valganciclovir or ganciclovir; however, treatment with these agents is limited by myelosuppression. In 2017, letermovir was approved by the FDA for CMV prophylaxis through day 100 post-HSCT in CMV-seropositive patients, later becoming a Grade A-1 recommendation by the ASTCT [24,65]. Cesaro et al. surveyed treatment approaches to CMV infection in a 2020 survey among European BMT centers, highlighting that up to 62% of centers are adopting a prophylaxis-based approach with letermovir [66]. Different strategies and the impact of letermovir prophylaxis on CMV reactivation are being studied [63,67,68].

Ganciclovir-resistant CMV is an emerging concern in the post-HSCT population, with treatment options including foscarnet, cidofovir, and maribavir. Notably, these antivirals have adverse effects: foscarnet and cidofovir can cause nephrotoxicity, while maribavir can interact with CYP3A4 substrates resulting in increased tacrolimus concentrations [69,70]. Publications by Jorgenson et al. describe a pharmacy-directed approach to CMV antiviral stewardship in SOT programs that has been associated with a decrease in the number of patients requiring treatment and a reduced ganciclovir resistance rate with a monitoring protocol for CMV viremia in patients receiving prophylaxis and treatment [71,72]. This type of antiviral stewardship approach may be an area of further exploration toward preventing CMV resistance in the post-HSCT population.

Other viral infections unique to oncology patients include acyclovir-resistant herpesvirus, BK polyoma virus, and adenovirus. Acyclovir (ACV)-resistant herpesvirus (HSV) has a prevalence of 3.5–10% among HSCT patients and can cause significant morbidity with prolonged viral shedding [73,74]. There are limited treatment options for ACV-resistant HSV, including foscarnet, cidofovir, and the investigative drug, pritelivir. Interventions, such as accessible diagnostic methods for resistance testing and improved dosing strategies for antiviral agents, may assist ASP in preventing antiviral resistance. Notably, no approved antiviral treatment options exist for BK polyoma virus or adenovirus with studies revealing a cautious benefit of cidofovir at the cost of nephrotoxicity; this example highlights another area requiring further research in terms of the identification of less harmful and more effective therapeutic approaches [67].

### 3.5. Adverse Effects of Antimicrobials

In addition to the adverse effects of antimicrobials, including systemic toxicity and DDI, antimicrobials can increase MDRO incidence through multiple mechanisms. As described previously, exposure to the healthcare system and antimicrobials can lead to the selection of and colonization with resistant organisms (e.g., ESBL organisms, CRE, VRE) [20,21]. Treatment centers should review resistance patterns and prevalence of MDRO at their institutions with consideration of ASP interventions to optimize the use of antimicrobials in an effort to prevent antimicrobial resistance [37,75,76].

Long-term antimicrobial use has also been linked to the disruption of the intestinal microbiome and an increased incidence of *C. difficile* infections in all populations [77]. *C. difficile* occurs with an increased incidence among patients with hematologic malignancies (up to 6.5 times higher) and HSCT (up to 9 times higher) when compared to the general inpatient population [78,79]. It is important to note that in addition to microbiota injury from antimicrobial exposure, HSCT recipients are also exposed to conditioning regimens and mucositis that can also disrupt the gut microbiota [80,81,82,83].

The disruption of the gut microbiome can result in serious, long-term consequences for patients living with malignancies. Galloway-Peña et al. observed that patients with a lower stool Shannon diversity index during initiation of chemotherapy for AML with resulting neutropenia experienced an increased infection incidence [84]. Lack of microbiome diversity during the peri-transplant period has been associated with poor overall survival post-HSCT and increased GVHD incidence [82,85,86]. Antimicrobial use after stem cell transplantation can compound this microbiota disruption and damage [87,88].

### 3.6. Expansion of Diagnostic and Susceptibility Testing Methods

The development of newer RDT platforms has made it possible for the earlier diagnosis of infections and for the determination of appropriate antimicrobial treatments. When compared to conventional microbiologic methods for pathogen identification, which may include subculture isolation, rapid molecular methods and matrix-assisted laser desorption/ionization–time-of-flight (MALDI-TOF) mass spectrometry offer quicker turn-around times in microbe identification [89,90,91]. Multiplex polymerase chain reaction (PCR) syndromic panels, such as panels developed for respiratory and gastrointestinal illnesses, allow for rapid diagnosis through the testing of multiple pathogens from a single sample.

Genotypic and phenotypic testing platforms performed on direct blood culture specimens (rather than on colonies isolated from subculture) have facilitated the rapid de-escalation of antibacterial agents and the early detection of MDRO for the potential escalation of therapy [92,93]. Platforms such as Accelerate Pheno^®^ can offer rapid pathogen identification results within 90 min, as well as report antimicrobial susceptibility test results within 7 h [93]. The T2Candida^®^ panel, which utilizes direct whole-blood specimens, can aid ASPs, both in the discontinuation of empiric antifungal use for candidemia due to its high negative predictive value for candidemia, as well as support the continuation of antifungals with a positive result due to increased sensitivity compared to blood cultures [54]. Local centers may consider the benefits of running RDT on-site and rapid turn-around time for results to optimize antimicrobial use, including the early identification of MDRO for use of extended-spectrum antimicrobials, as well as assistance with antibiotic de-escalation.

Another diagnostic challenge is the diagnosis of IFI. Currently, the sensitivity of serum *Aspergillus* galactomannan is low in patients receiving mold-active antifungals, which may limit its use in diagnosing IFI in patients receiving antifungal prophylaxis [54,55]. Further studies regarding the use of indirect testing methods for the diagnosis of IFI, as well ongoing evaluations of diagnostic definitions, may assist clinicians in diagnosing IFI and might assist ASPs in optimizing antifungal use.

### 3.7. Lack of Dedicated ASP Guidelines and the Need for Multidisciplinary Implementation

In 2016, the Infectious Diseases Society of America (IDSA) and the Society for Healthcare Epidemiology of America (SHEA) created a joint guideline regarding general ASP implementation in hospitals. Within these guidelines, specific recommendations regarding HSCT and oncology patients are made, suggesting the development of facility-specific guidelines for fever and neutropenia, as well as efforts towards antifungal stewardship, including the incorporation of non-invasive fungal markers to optimize antifungal use [60]. Currently, no consensus society guidelines exist regarding the implementation of ASPs specifically targeting immunocompromised patients with malignancies.

Emerging data continue to note the importance of different stewardship-related interventions in HSCT and oncology patients; however, the consistency of stewardship implementation has varied across the centers who manage these patients [6,7]. An Australian survey of antimicrobial prescription practices highlights differences in the adherence to guidelines for febrile neutropenia and antimicrobial use among BMT and other oncology services, emphasizing the need for the development of local guidelines [94]. In a survey done by Seo et al., approximately 76% of respondents from facilities who performed HSCT in the United States were involved in local guideline developments that were unique to the oncology population, such as those targeting febrile neutropenia; however, only 34% of the centers tracked outcomes related to antimicrobial use for these patients [6].

Figure 2 outlines a proposed multidisciplinary team approach towards ASP implementation to optimize educational opportunities for providers and to optimize patient outcomes. Elements in this model include stakeholder input from the patient and/or caregiver, oncology provider, infectious disease provider, pharmacists, and nurses, with ongoing input from the infection prevention department and microbiology laboratories. It is important that these teams work collaboratively within the lens of reporting, quality, and safety in order to create a comprehensive approach regarding antimicrobial use.

### 3.8. Summary of Opportunities for ASP Interventions Specific to Oncology and HSCT Patients

In 2019, the CDC outlined core elements of hospital antibiotic stewardship programs, which included leadership commitment, accountability, pharmacy expertise, action, tracking, reporting, and education. In Table 1, we summarize some ASP interventions previously discussed in this article and described in other studies within this framework as areas of focus, improvement, and future research regarding oncology and HSCT patients [6,95].

## 4. Conclusions

Patients living with solid tumors and hematologic malignancies are at an increased risk of infection due to underlying malignancies, immune dysfunction due to therapeutic agents, and other unique aspects of care. Many advances have occurred regarding expanded treatment options for solid tumors and hematologic malignancies; however, emerging data suggest that some of these newer therapies are associated with an increased risk for opportunistic infections. Studies have supported antimicrobial strategies to decrease the incidence and morbidity associated with opportunistic infections in these immunocompromised populations, such as antimicrobial prophylaxis; yet, the implementation of these strategies remains inconsistent.

More research is needed pertaining to the optimal use and development of novel antimicrobial agents for infections that are unique to the HSCT and oncologic populations, including antibacterial treatment for MDRO infections, antifungals for prophylaxis and treatment of resistant IFI, and the antiviral treatment of resistant CMV and HSV. Downstream consequences of different antimicrobial agents, including the development of resistant pathogens, disruption of the host microbiome, and long-term morbidity and mortality, are poorly described. Ongoing advances in RDT aimed at early pathogen detection and resistance genotype and phenotype determination will help ASPs and clinicians optimize antimicrobial use.

Given the complex components required to optimally care for patients undergoing HSCT and patients who are being treated for underlying malignancy, a multidisciplinary approach is imperative in order to successfully implement ASP and its interventions. It is important for institutions that care for these high-risk populations to implement collaborative ASP strategies to optimize antimicrobial use and to deliver safe outcomes in patients living with malignancies.

## Figures and Tables

**Figure 1 antibiotics-12-00592-f001:**
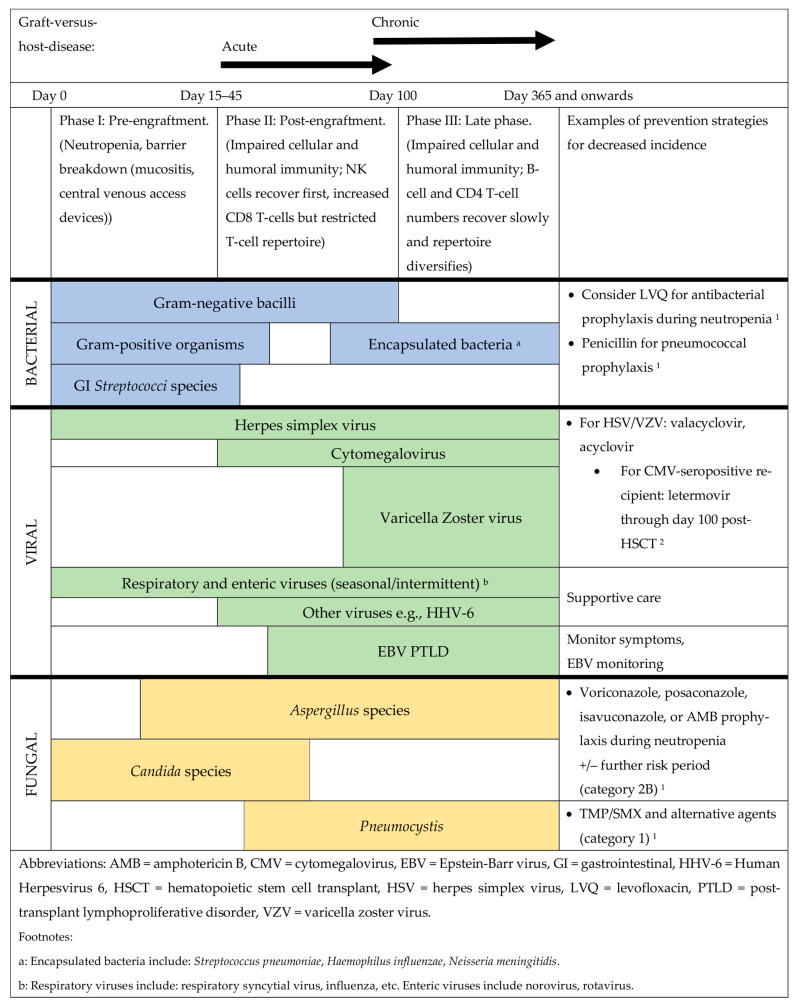
Phases of opportunistic infections among allogeneic HSCT recipients and examples of prevention strategies—Adapted with permission from Ref. [23]. Copyright 2016, Elsevier. ^1^ Reference [11]. ^2^ Reference [24].

**Figure 2 antibiotics-12-00592-f002:**
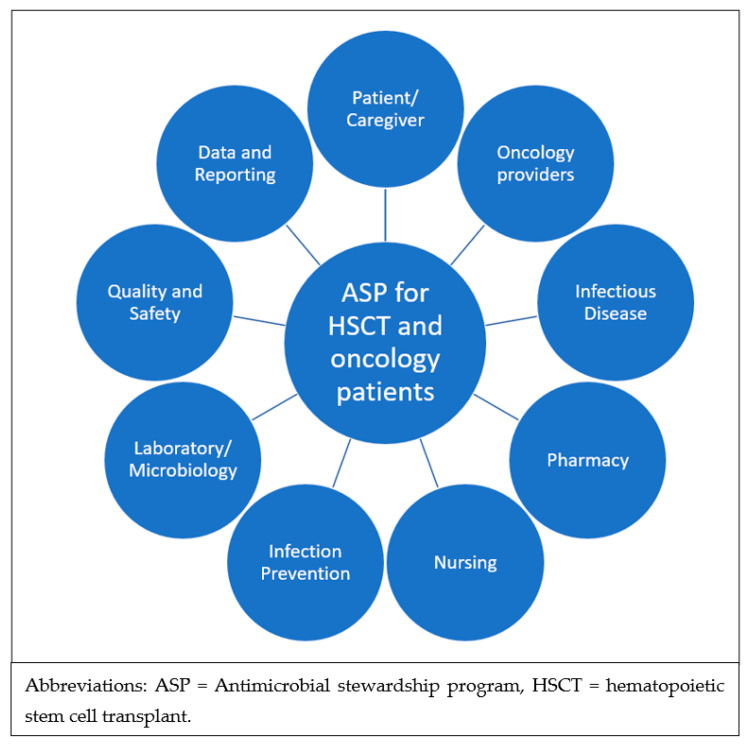
Multidisciplinary approach for ASP interventions in HSCT and oncology patients—Adapted with permission from Ref. [4]. Copyright 2022, Elsevier.

**Table 1 antibiotics-12-00592-t001:** Examples of ASP Interventions Specific to Oncology Patients Aligned with CDC Core Elements.

CDC Core Element	Sample Interventions Focused on HSCT/Oncology Patients
**Hospital Leadership Commitment** Dedicate necessary human, financial, and information technology resources.	Accessible information systems (e.g., electronic medical record, surveillance data) Dedicated staff for antimicrobial stewardship
**Accountability** Appoint a leader or co-leaders, such as a physician and pharmacist, responsible for program management and outcomes.	Multidisciplinary approach among hematology/oncology, infectious disease, and pharmacy (“handshake stewardship”)
**Pharmacy Expertise** Appoint a pharmacist, ideally as the co-leader of the stewardship program, to lead implementation efforts to improve antibiotic use.	Antibacterial, antifungal, and antiviral prophylaxis Dose optimization (e.g., extended infusion of beta-lactams) Duration of empiric antimicrobials for febrile neutropenia IV to PO conversion
**Action** Implement interventions, such as prospective auditing and feedback, or preauthorization, to improve antibiotic use.	Development of population specific guidelines Febrile neutropenia Antifungal prophylaxis and treatment Cytomegalovirus prophylaxis Use of microbiology methods to assist with prescribing
**Tracking** Monitor antibiotic prescribing, impact of interventions, and other important outcomes such as *C. difficile* infection and resistance patterns.	Population- and/or unit-specific antibiograms Prevalence of MDRO Prospective audit and formulary restriction
**Reporting** Regularly report information on antibiotic use and resistance to prescribers, pharmacists, nurses, and hospital leadership.	Tracking and shared reporting of outcomes specific to HSCT/oncology *C. difficile* Catheter-related infections Prevalence of MDRO
**Education** Educate prescribers, pharmacists, and nurses about adverse reactions to antibiotics, antibiotic resistance, and optimal prescribing.	Population-specific antibiograms Microbiome diversity

Abbreviations: ASP = antimicrobial stewardship program, CDC = Centers for Disease Control and Prevention, *C. difficile* = *Clostridioides difficile*, HSCT = hematopoietic stem cell transplant, IV = intravenous, MDRO = multi-drug resistant organism, PO = by mouth.

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
