# Peer review of "Challenges and Opportunities in Antimicrobial Stewardship among Hematopoietic Stem Cell Transplant and Oncology Patients"

_antibiotics, 2023, doi:10.3390/antibiotics12030592_

Round 1

Reviewer 1 Report

 I found this article interesting for the readers and followed the journal’s scope. I don’t have any major comments as this prospective article has enough data and is well written with proper discussion.

I would recommend the article be published in Antibiotics after minor corrections. 

The author needs to address the following comments/corrections.

1.     The author should correct reference # 2.

2.     Figure 1, 2 and table 1 need footnotes and need to be discussed a bit.

3.     The author could include figures or table for the discussion for 2.2, 3.1, 3.2, 3.3, 3.4, 3.5, 3.6 and 3.7 (e.g. treatment options, available therapies, consequences etc.).

4.     As Figures 1 and 2 and Table 1 are already included in the manuscript, therefore, supporting document having these data are unnecessary.

Author Response

I found this article interesting for the readers and followed the journal’s scope. I don’t have any major comments as this prospective article has enough data and is well written with proper discussion.

I would recommend the article be published in Antibiotics after minor corrections. 

The author needs to address the following comments/corrections.

  1. The author should correct reference # 2.
    Thank you for your feedback. The authors agree and Reference #2 has been updated and corrected.  
  2. Figure 1, 2 and table 1 need footnotes and need to be discussed a bit.
    Thank you for this suggestion Figure 1 has been revised with a few footnotes to elaborate on certain encapsulated bacteria and viruses. Figure 1 also has expanded in-text discussion within lines 92 through 94. Figure 2 has been discussed within text discussion in lines 365 through 369. Table 1 has further in-text discussion in lines 379-380. Table 1 is a summary of possible interventions mentioned in previous sections so further elaboration within text seemed redundant; however the authors are open to suggestions. The authors would appreciate any clarification from the reviewer what would be desired by a footnote in Figure 2 and Table 1, if applicable. 
  3. The author could include figures or table for the discussion for 2.2, 3.1, 3.2, 3.3, 3.4, 3.5, 3.6 and 3.7 (e.g. treatment options, available therapies, consequences etc.).
    Thank you for this feedback. Although the authors agree with the reviewer’s comment regarding inclusion of further figures and tables within these sections, the authors feel that addition of some of these suggestions may detract from the nature of this perspective piece mainly focused on antimicrobial stewardship.
  4. As Figures 1 and 2 and Table 1 are already included in the manuscript, therefore, supporting document having these data are unnecessary.
    Thank you for notifying us. This suggestion has been noted by the authors. 

Reviewer 2 Report

The article was well-written, with the notion of antimicrobials being expanded to include antifungal and antiviral drugs as well.

The statement “…poor outcomes have informed therapeutic decisions…” on line 18 is confusing. Did the authors actually mean 'influenced' instead of 'informed'?

It is unclear what the 'destruction' on line 46 refers to. Please specify.

In Figure 1, some explanation on encapsulated bacteria is needed because of the overlap with both gram-positive and gram-negative bacteria. For example, Klebsiella pneumoniae is both an encapsulated bacterium as well as a gram-negative rod and pneumococci are encapsulated as well as gram-positive. Probably specific bacterial names should be mentioned.

Likewise, in Fig 1, specific virus names can be mentioned for 'respiratory and enteric viruses'

'Enterobacterales' is an order and not a genus. Thus, it should not be italicized on line 165.

Author Response

The article was well-written, with the notion of antimicrobials being expanded to include antifungal and antiviral drugs as well.

The statement “…poor outcomes have informed therapeutic decisions…” on line 18 is confusing. Did the authors actually mean 'influenced' instead of 'informed'?

Thank you for your feedback. The word “informed” has been replaced with “shaped” in line 18 of revised manuscript.

It is unclear what the 'destruction' on line 46 refers to. Please specify.

We have added further language and elaboration around the word “destruction” in lines 45-47. 

In Figure 1, some explanation on encapsulated bacteria is needed because of the overlap with both gram-positive and gram-negative bacteria. For example, Klebsiella pneumoniae is both an encapsulated bacterium as well as a gram-negative rod and pneumococci are encapsulated as well as gram-positive. Probably specific bacterial names should be mentioned.

The authors appreciate this suggestion. We have provided additional footnotes and examples for “encapsulated bacteria” in Table 1.  

Likewise, in Fig 1, specific virus names can be mentioned for 'respiratory and enteric viruses'

Additionally, we have provided footnotes and examples for “enteric viruses” and “respiratory viruses” in Table 1. 

'Enterobacterales' is an order and not a genus. Thus, it should not be italicized on line 165.

Thank you for notifying us. We agree with the reviewer and the word is no longer italicized. 

Reviewer 3 Report

The study well designed and drafted. Can be published in the present form.

Author Response

We appreciate and thank you for your feedback.